# IMPROVING HUMAN POSE-CONDITIONED GENERATION: FINE-TUNING CONTROLNET MODELS WITH REINFORCEMENT LEARNING

## ABSTRACT

Advancements in diffusion-based text-to-image generation models have made it possible to create high-quality human images. However, generating humans in desired poses using text prompts alone remains challenging. Image-to-image generation methods utilizing additional image conditions can address this issue; however, they often struggle with generating images that accurately match conditioning images. This paper proposes a new fine-tuning framework for training ControlNet models with reinforcement learning by combining ControlNet and Denoising Diffusion Policy Optimization (DDPO) to understand pose conditioning images better. We apply a novel reward function in the proposed framework to increase pose accuracy. We demonstrate that our method effectively improves human generation by enhancing pose accuracy and the correct generation of body parts without omissions or additions. Furthermore, we prove that the usage of more detailed pose dataset along with our proposed reward functions leads to improved training results.

## 1 INTRODUCTION

Advancements in text-to-image generation models, such as Stable Diffusion, (Rombach et al., 2022; Croitoru et al., 2023; Cao et al., 2024; Zhang et al., 2023a) have enabled the generation of high-quality human images. However, generating humans in desired poses using text input alone is challenging. An image-to-image generation method is employed, utilizing additional image conditions, such as depth map images(Eigen et al., 2014) or OpenPose images(Cao et al., 2017). This allows new images to be generated based on the content of given conditions, enabling humans to create desired poses.

Additional model training is necessary to accept new image conditions as input. While it is possible to directly fine-tune the large generative model, this approach requires expensive computing resources and large datasets. More efficient methods such as ControlNet(Zhang et al., 2023b) and T2I-Adapters(Mou et al., 2024) have been proposed to address these limitations. These methods enable learning new conditioning images with limited data and computing resources.

Utilizing these approaches allows for controlling the structural aspects during image generation, making it much easier to generate humans in desired poses. However, even when using conditioning images, the models often struggle to create complex poses and frequently fail to generate images that perfectly match the conditioning image pose.

In this paper, we propose a new fine-tuning framework that combines ControlNet(Zhang et al., 2023b) and Denoising Diffusion Policy Optimization (DDPO)(Black et al., 2023) that can train models to understand conditioning images better; Refer to Figure 1. DDPO is a diffusion-based reinforcement learning method that trains the diffusion model to perform better on specific tasks like text-to-image alignment and aesthetic quality. In the proposed framework, ControlNet approach of freezing the existing model and training only the copied neural network to learn new conditions is combined with DDPO approach of applying reinforcement learning to the denoising procedure of the diffusion model.

Figure 1: Proposed Architecture - The human pose-conditioning image and the text prompt are the inputs to ControlNet diffusion model. The novel reward function further improves human pose alignment.

Our contributions are as follows:

1. We propose a novel fine-tuning framework that enhances the reflection of conditioning images by integrating reinforcement learning into ControlNet models (Zhang et al., 2023b).

2. We introduce novel reward functions suitable for training on human pose conditions, resulting in higher pose accuracy than the baseline models.

3. We demonstrate the effectiveness of using more detailed pose dataset and our reward functions, leading to improved training outcomes.

## 2 RELATED WORK

### 2.1 DENOISING DIFFUSION MODELS

Denoising diffusion models (Ho et al., 2020) have become a powerful class of generative models. These approaches have been extended to various domains, including audio (Schneider, 2023), text generation (Li et al., 2022), and text-to-image generation (Rombach et al., 2022). More recently, diffusion models have emerged that allow customization of the output image by entering a detailed description as a conditioning input rather than simply generating the image. Conditioning inputs can take many different forms and be used to control various aspects. For example, DALLE2 (Ramesh et al., 2022) uses text as a condition for image generation, GLIGEN (Li et al., 2023) controls position, and PIDM (Bhunia et al., 2023) is used to control pose. In this paper, our conditioning input is the pose image extracted from the original image.

### 2.2 POSE-GUIDED HUMAN IMAGE GENERATION.

The development of pose-guided human image generation starts with GAN-based methods (Mirza & Osindero, 2014) and advances to diffusion models. Early approaches, like those by Ma et al. (2017), used conditional GANs but struggled with pose misalignment. To address this, Esser et al. (2018) introduced VUnet, combining VAEs (Kingma & Welling, 2013) and U-Net (Ronneberger et al., 2015) architectures to better decouple pose and appearance. Def-GAN by Siarohin et al. (2018) further improved alignment. Subsequently, GFLA by Ren et al. (2020), employed flow-based deformations to improve pose alignment performance, while models like ClothFlow (Han et al., 2019) used segmentation maps for guidance. Attention-based approaches like CoCosNet (Zhou et al., 2021) extracted dense correspondences, enhancing texture and spatial consistency. Recently, diffusion models like PoCoLD (Han et al., 2023) and PIDM (Bhunia et al., 2023) have emerged, using cross-attention mechanisms to model spatial correspondences, overcoming the limitations of GANs (Goodfellow

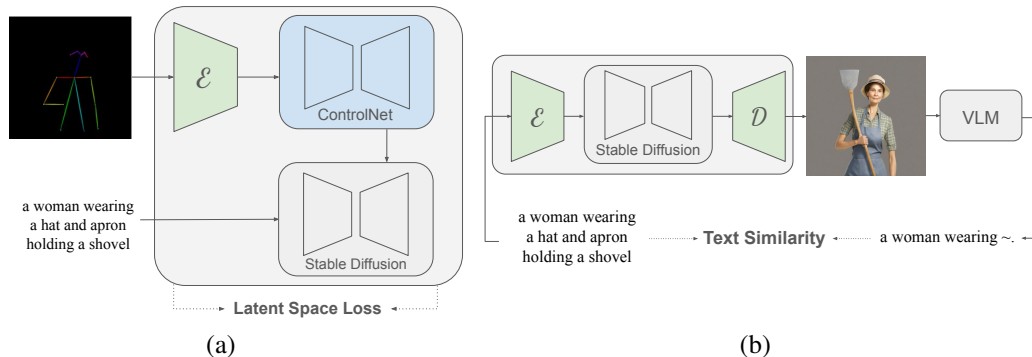

Figure 2: Training Architectures of ControlNet and DDPO. (a) ControlNet Architecture (Zhang et al., 2023b). The conditioning image is a human pose image. (b) DDPO Architecture (Black et al., 2023), the text similarity reward function further improves text-image alignment.

et al., 2020), such as instability while training and difficulty in high-resolution synthesis. These advancements have led to more realistic and high-quality pose-guided human image generation.

However, the above-mentioned methods do not generate a completely new person but rather transform the person in the source image into a desired pose. To create entirely new images based on poses, one can generate images using text and pose images as conditions instead of a source image. The diffusion model must be trained on new conditioning images for this to work. Methods such as ControlNet(Zhang et al., 2023b) can be used to train a large diffusion model for this purpose. Uni-ControlNet further extends this by allowing the model to handle multiple conditioning inputs via a unified adapter structure, improving flexibility and accuracy (Zhao et al., 2024). HumanSD improves pose conditioning by directly training diffusion models with heatmap-guided denoising loss (Ju et al., 2023). Stable-Pose advances the pose-conditioned image generation by incorporating Vision Transformers (ViT)(Dosovitskiy, 2020) to capture patch-level relationships in poses, enabling a deeper understanding of pose information (Wang et al., 2024).

However, models trained in these ways still sometimes fail to generate images correctly based on pose conditions, for example, by creating additional joints or failing to accurately replicate the pose in the conditioning image. To improve the model's accuracy regarding pose conditions, in this paper, we propose a method to further train ControlNet using reinforcement learning.

### 2.2.1 ADDING CONDITIONAL CONTROL TO TEXT-TO-IMAGE DIFFUSION MODELS (CONTROLNET)

ControlNet(Zhang et al., 2023b) is a neural network architecture that allows diffusion models to learn new additional conditions for precise structural adjustments during image generation. ControlNet has a structure where a neural network from large models, like Stable diffusion(Rombach et al., 2022), is duplicated and connected to the original block. The existing network remains untrained, while only the newly connected network undergoes, as depicted in Figure 2 (a). Consequently, this allows for maintaining the performance of the existing large model while learning new conditions. It also has the advantage of being trainable with fewer computing resources on a smaller dataset than directly fine-tuning Stable Diffusion.

ControlNet(Zhang et al., 2023b) is currently used for various tasks, one of which is training pose images. It is useful for generating human images in desired poses because generating a person in a desired pose using only text input is challenging. However, it still has limitations in generating detailed parts of human body or fails to generate complex poses. This paper aims to apply Reinforcement learning-based approaches to Controlnet to better reflect pose conditioning images.

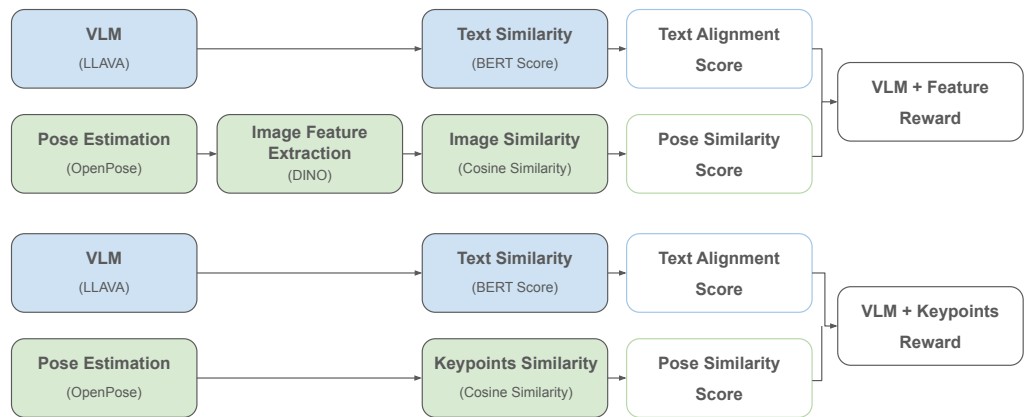

Figure 3: Proposed Reward Functions composed of image-text alignment components and pose similarity base components.

## 2.3 REINFORCEMENT LEARNING AND DIFFUSION MODELS.

Bachman & Precup (2015) assumed data generation as a sequential decision-making problem preceding the diffusion model and introduced reinforcement learning methods to image generation using their framework. Furthermore, Fan & Lee (2023) applied policy gradient methods to improve data distribution matching. As the objective is not to optimize downstream tasks, they considered the reward function as a discriminator of GAN architecture(Goodfellow et al., 2020).

In concurrent works, Diffusion Policy Optimization with KL regularization (DPOK)(Fan et al., 2024) and DDPO(Black et al., 2023) employed RL in the diffusion process. DPOK attempted to improve human preferences using policy gradient methods with KL regularization and trained different models on each prompt. DDPO, in contrast, excluded KL regularization, especially for text alignment enhancement, which is one of the objectives they addressed, harnessed the describing ability of Vision-Language Models (VLM), and trained the models on many prompts at once.

We built upon DDPO (Black et al., 2023) targeting better reflection of conditioning image, following the same training condition as DDPO, using VLM instead of adopting KL-regularization and training on many prompts simultaneously. However, we introduced a new reward function to enhance the pose accuracy simultaneously while still adopting VLM-reward function to maintain the text-to-image diffusion models' text alignment ability.

### 2.3.1 DENOISING DIFFUSION POLICY OPTIMIZATION (DDPO)

DDPO(Black et al., 2023) is a reinforcement learning method for denoising diffusion models. Its objective is to maximize the reward of samples and contexts induced by

$p_\theta(x_0|c)$ and $p(c)$:

$$\mathcal{J}(\theta) = \mathbb{E}_{c \sim p(c), x_0 \sim p_\theta(x_0|c)}[r(x_0, c)]. \tag{1}$$

DDPO(Black et al., 2023) applies a multi-step MDP, treating the iterative denoising procedure as a Markov Decision Process (MDP). It maximizes $\mathcal{J}(\theta)$ by using policy gradient estimation, a direct Monte Carlo estimate with likelihoods and its gradients. An importance sampling estimator is added to the equation for the trust region clipping.

DDPO(Black et al., 2023) proves their performance by evaluating their method on text-to-image diffusion with various reward functions. Figure 2 (b) describes the pipeline of DDPO applied to enhance text alignment performance. In this pipeline, they propose VLM-reward function, which leverages VLM's ability to understand and explain visual information in text and is used to improve text alignment. VLM in the reward function takes the image generated by the diffusion model and the text prompt, asking, "What is happening in this image?". As the texts generated by VLM

| Prompt | Image | Openpose w/o hands and face | Openpose w/ hands and face |
|--------|-------|-----------------------------|----------------------------|
| A young Asian woman, professional business entrepreneur in a brown suit with arms crossed and smile isolated over white background |  |  |  |

Table 1: Train Dataset Components- Each training sample contains an image, a text-based description of the image, and OpenPose skeleton of the image with and without hands and face keypoints.

describe the general idea of the generated image, the reward based on the similarity between the intention of the prompt for the diffusion model and the generated image can be yielded by using BERTScore(Zhang et al., 2019). Likewise, DDPO can adapt a pre-trained model by defining reward functions.

DDPO(Black et al., 2023) is not designed for learning new conditioning images. In this paper, we overcome this restriction by adopting ControlNet model(Zhang et al., 2023b) to DDPO training architecture.

## 3 METHODOLOGY

This paper proposes a fine-tuning framework for image-conditioned text-to-image models to enhance the model's ability to more accurately generate human images conditioned on various poses. In the subsequent sections, we will explain the details of the model's architecture and reward functions.

### 3.1 THE PROPOSED ARCHITECTURE

Our proposed framework fine-tunes ControlNet(Zhang et al., 2023b) with the DDPO(Black et al., 2023) strategy. The original DDPO applies reinforcement learning to Stable Diffusion model. However, we replace this with ControlNet model to enable learning new conditioning images. Since ControlNet uses latent space loss for training, we modified this to a reward system and applied DDPO's multi-step MDP. Additionally, unlike DDPO, we focus on pose-based human image generation, so we use a pose accuracy reward in addition to DDPO's text alignment reward. As mentioned before, the overall architecture of our model is depicted in Figure 1.

In the fine-tuning procedure, the first step involves feeding the OpenPose conditioning image(Cao et al., 2017) and the text prompts into the model as inputs, generating an image as output. Next, each component of the reward function produces individual rewards. Finally, the rewards are aggregated, and the total reward is returned to the model during denoising.

### 3.2 PROPOSED REWARD FUNCTIONS

One crucial aspect of our fine-tuning framework is our choice of reward model, crafted to maintain image-text alignment and improve image-human pose alignment. We propose two kinds of reward functions:

- VLM + Feature Reward Function

- VLM + Keypoints Reward Function

| Model | Type of Condition | Type of Reward |
|---|---|---|
| ControlNet | OpenPose w & w/o hands&face | NA |
| Uni-ControlNet | OpenPose w & w/o hands&face | NA |
| StablePose | MMPose | NA |
| HumanSD | MMPose | NA |
| **Model 1 (ours)** | OpenPose w/o hands&face | VLM+Feature Reward |
| **Model 2 (ours)** | OpenPose w/o hands&face | VLM+KeyPoints Reward |
| **Model 3 (ours)** | OpenPose w hands&face | VLM+Feature Reward |
| **Model 4 (ours)** | OpenPose w hands&face | VLM+KeyPoints Reward |

Table 2: Models for evaluations - Four baselines (ControlNet, Uni-ControlNet, StablePose, HumanSD) are compared with our models.

We adopted DDPO's VLM-reward function for text alignment as shown in Figure 4, utilizing LLaVA (Liu et al., 2023) as the VLM and BERTScore (Zhang et al., 2019) as the reward score. VLM receives the generated image and a text prompt asking, "What is shown in this image?". LLaVA works as an image-captioning model that can analyze the image and provide a clear description in natural language that accurately reflects its content and context. The BERTScore between the description and inputted text prompt to ControlNet model is calculated and used for the text alignment score.

We propose two variants of the reward function for pose accuracy: Feature reward function and Keypoints reward function. The pose estimation model(Openpose) (Cao et al., 2017) is required for both variants to extract OpenPose images or coordinate of the keypoints from the generated image.

Feature reward function yields the rewards based on the cosine similarity of features extracted from two images: an input conditioning image and an OpenPose image extracted from a generated image; thus, it requires a feature extraction model, in which we used DinoV2-base model (Oquab et al., 2024). In order to focus only on a pose in images, we do not compare a generated image with an original image, but rather compare OpenPose images of those two. On the other hand, Keypoints reward function works based on the cosine similarity between the keypoints of the joints from the input conditioning image and OpenPose of the generated image.

## 4 EXPERIMENTAL SETUP AND RESULTS

### 4.1 DATASET

We prepared 21K pose related images and alternative text pairs by collecting them from several license-free image sites (Freepik; AdobeStock) and leveraging some parts of testset of Open Image v7 dataset (Kuznetsova et al., 2020). We ensured that the images contained a human throughout the collection process. The resolution of collected images is $1024 \times 1024$. Both types of OpenPose images(Cao et al., 2017), with or without facial landmarks and hands keypoints, are extracted from each image. Each dataset sample consists of three images and a text component: the collected image, OpenPose image, OpenPose with hands and face keypoints, and image caption. The dataset is split into train and test at a ratio of 20 to 1. The test dataset consists of 1K real images of a single person.

### 4.2 EVALUATION STRATEGY

We compared our models with four baseline models, ControlNet, Uni-ControlNet, StablePose, and HumanSD. Two different types of evaluations are conducted: Evaluation 1 focuses on the base model used for training, while Evaluation 2 focuses on other baseline models. The purpose of Evaluation 1 is to first assess whether our framework improves pose accuracy, and second, to examine the impact of using more detailed poses including hands and face. Evaluation 2 aims to compare our model's performance against other pose-based human image generation models, regardless of condition types. In both types of evaluation, three main criteria on which we focused to evaluate the

quality of generated images: pose accuracy, image quality, and text-image alignment. Improvement in pose accuracy ensures our framework is effective, and therefore, our primary goal of evaluation is to enhance pose accuracy. However, image quality and text-image alignment are also important components in image generation, so we additionally assess those two criteria to evaluate whether our models improve the pose accuracy without degradation of their current ability.

**Pose Accuracy:** We use Object Keypoint Similarity (OKS) (Maji et al., 2022) as our primary metric for assessing pose accuracy, measuring the distance between similarity in original and generated images on a scale from 0 to 1. A higher OKS score indicates better pose accuracy. Our objective is to improve OKS score compared to the baseline models.

**Image Quality:** To assess image quality, we utilize LPIPS (Learned Perceptual Image Patch Similarity)(Zhang et al., 2018), which measures perceptual similarity between the original and generated images, with scores ranging from 0 to 1. Lower LPIPS scores indicate better image quality and higher similarity to the original images. This measure has been demonstrated to closely match human perception. Our goal is to maintain LPIPS score of the baseline models.

**Text Alignment:** CLIP score(Hessel et al., 2021) is used for text alignment, which evaluates the similarity between generated images and their corresponding text representations, ranging from 1 to 100. Higher CLIP scores signify greater alignment between the text and the image. Like LPIPS score, CLIP score correlates highly with human judgment. Our objective is to preserve CLIP score of the baseline models.

## 4.3 TRAINED MODELS USING THE PROPOSED FRAMEWORK

The base model for training is Controlnet+OpenPose model(con) trained on Stable Diffusion XL (Podell et al., 2023). Four models are defined to evaluate the proposed frame comprehensively, each trained with different combinations of datasets and types of rewards.

These models are distinguished by the types of OpenPose conditioning images and rewards used during training. The first and second models are trained using OpenPose images without hands and face keypoints as conditions. In contrast, the third and fourth models utilized more detailed Open-Pose images, including hands and face keypoints. Models 1 and 3 employ the first reward function, VLM + feature reward function. Conversely, models 2 and 4 are trained using the second type of reward function, VLM + keypoints reward function. The choice of these models enables us to perform detailed ablation studies on the effect of different reward models and the conditioning images' details. We expect to be able to generate more detailed human pose images incorporating facial landmarks and hands keypoints and better pose alignment through our proposed reward models.

**Training Details:** In addition, our models are trained with four NVIDIA RTX A6000 using the same conditions: learning rate = $2 \times 10^{-6}$, batch size per GPU is two images, and sampling steps of 50. Training steps for models 1 to 4 are 30025, 18775, 50525, and 19025, respectively. It varies as we use the model which is saved when the reward is maximized before the models are overfitted.

## 4.4 QUANTITATIVE RESULTS

The quantitative evaluation results of all the models, along with the training base model (Control-Net), are shown in Table 3. In both condition types, the proposed trained models outperform the base model in terms of pose accuracy, which proves the effect of our reward functions regardless of the types of condition. In addition, models 3 and 4, trained with OpenPose images that include hands and face, show superior performance compared to the model trained on OpenPose images without hands and face. This demonstrates that containing more detailed input images gives positive impact on pose-guided human generation. In the case of CLIP Score and LPIPS, considering their score range, our models stay in similar level compared to the base model, which means we successfully maintain their existing ability related to image quality and text alignment while improving pose accuracy.

There are results of Quantitative Evaluation 2 in Table 4. We compare our model with the highest score to the baseline models. In this case, our model shows the highest score except CLIP Score.

| Condition Type | Model | CLIP Score ↑ | LPIPS ↓ | OKS ↑ |
|---|---|---|---|---|
| OpenPose w/o hands and face | ControlNet | 29.2268 | 0.7236 | 0.5028 |
| | **Model 1 (ours)** | 29.0550 | 0.7230 | 0.5420 |
| | **Model 2 (ours)** | 28.8678 | 0.7269 | 0.6006 |
| OpenPose w/ hands and face | ControlNet | 29.1303 | 0.7137 | 0.6046 |
| | **Model 3 (ours)** | 28.8470 | 0.7257 | 0.6837 |
| | **Model 4 (ours)** | 28.9534 | 0.7167 | 0.6551 |

Table 3: Quantitative Evaluation 1. Quantitative evaluation with the base model for training. The best values in each metric and condition type are underlined.

| Condition Type | Model | CLIP Score ↑ | LPIPS ↓ | OKS ↑ |
|---|---|---|---|---|
| MMPose | HumanSD | 25.5067 | 0.7495 | 0.6574 |
| | StablePose | 18.9854 | 0.7558 | 0.6781 |
| OpenPose | Uni-ControlNet | 29.2736 | 0.7891 | 0.4849 |
| | **Model 3 (ours)** | 28.8470 | 0.7257 | 0.6837 |

Table 4: Quantitative Evaluation 2. We evaluate our model with the highest score with HumanSD, StablePose, and Uni-Controlnet. HumansSD and StablePose use MMPose and Uni-Controlnet uses OpenPose as a conditioning image. The best values in each metric are underlined.

Despite the best performance of our model with respect to pose accuracy, we expect enhancement from other models if our framework is applied as well. This is further discussed in section 5.

## 4.5 QUALITATIVE RESULTS

Figure 4 shows the generated images in different poses-conditioning images between the baseline model (ControlNet) and our Model 2. The results show our model more accurately generates poses without generating additional humans or body parts while maintaining image quality. For instance, in Fig. 4-column(a), the baseline model fails to generate the correct pose of the hand holding the shovel. The conditioning image shows a 90-degree angle, which is generated correctly by the proposed model. The arm posture of Fig. 4-column(e) shows a similar issue. The baseline model does not generate the image fully aligned with the arm position. In contrast, the proposed model could generate the arm upward, similar to the conditioning image. As shown for the little girl and boy images, Fig. 4-columns(c-d), not all the details of the conditioning image are captured by the baseline model. In contrast, our proposed reward function, which considers the pose similarity, fine-tuned the model to produce more aligned images with the conditioning images. The Fig. 4-column(b) shows an interesting observation, in which the conditioning image shows only one pose, and the prompt also talks about one woman standing with her arms crossed. Still, the baseline model generated two women with such a pose. Although both women follow the requested pose structure, the generated image is not well aligned with the prompt or the conditioning image. Our fine-tuning strategy and improvement of the pose alignment of the generated and conditioning images helped generate these samples, which are fully aligned with the conditioning images.

Figure 5 shows the conditioning image with additional face and hands keypoints. The baseline model (ControlNet) and our proposed Model 4 are used to generate the images. Model 4, as well as Model 3, went through the fine-tuning process with more detailed conditioning images. The additional hands and face keypoints add more constraints in the generation process, and the keypoint rewards model ensures a tight alignment between the pose of the generated image and the conditioning image. For instance, in Fig. 5-column(a) and column(b), Model 4 generates body parts including hands in more details than the baseline model does. Especially in Fig. 5-column(b), Model 4 generates not only the clear hands but also the accurate pose and perfect body parts. Furthermore, in

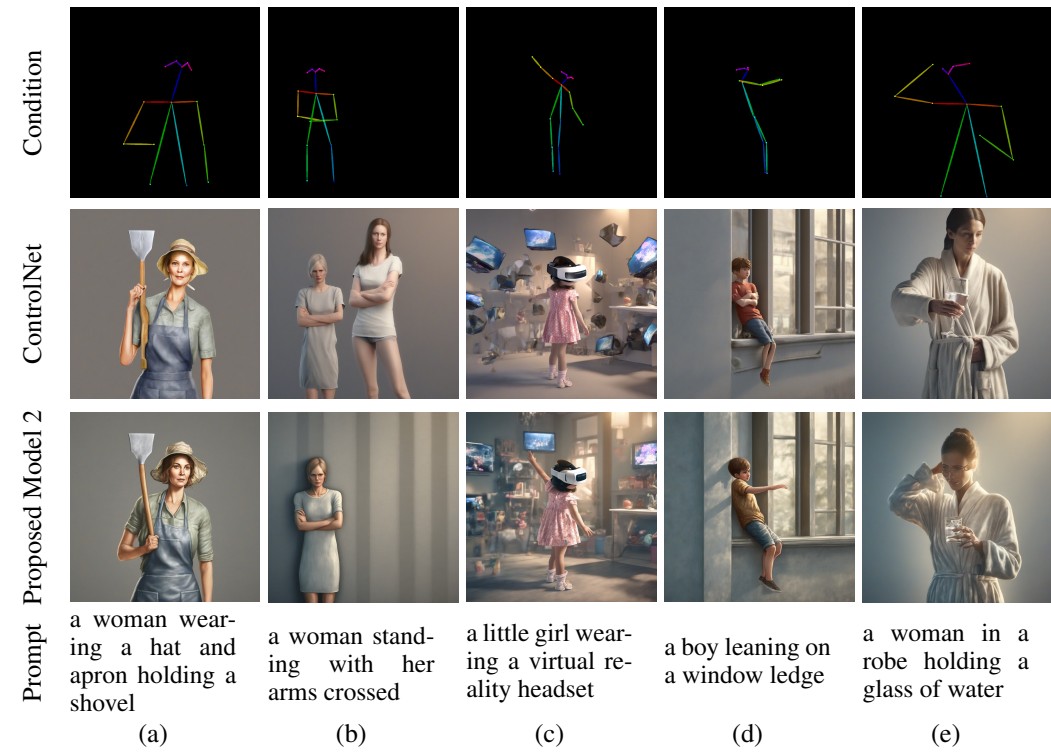

Figure 4: Qualitative Results. (first row) Skeleton image conditions, (second row) the generated images by ControlNet as a baseline model, (third row) the generated images by the proposed model 2, (last row) the input prompts.

Fig. 5-column(c-e), Model 4 generates images correctly following the conditioning images, such as the formation of legs or direction of arms, while the baseline model fails to follow them accurately.

Qualitative comparison between all baseline models and our models are shown in Appendix B; Figure 8 and Figure 9.

## 5 DISCUSSION AND FUTURE WORKS

Despite the significant impact of the reward function of the keypoints, we only employed it in the case of single-person training. This will be expanded to accommodate multi-person training by enhancing the reward function for the multi-person version. Additionally, we will explore ways to utilize resource-efficient strategies to minimize the cost of fine-tuning. More studies on the effect of quantization and smaller diffusion-based models would be an exciting research direction to democratize these approaches further and reduce the cost of fine-tuning in the proposed framework. Lastly, we will expand the application of the proposed fine-tuning framework to other models and expect substantial enhancement in terms of the pose accuracy in other cases as well. This is because our framework and two reward functions can be applied to other models utilizing any kinds of pose conditioning images including OpenPose and MMPose. Although our model outperforms other baseline models, the expansion of the application to the baseline models with higher pose accuracy, such as HumanSD and StablePose, will result in better performance.

## 6 CONCLUSION

We suggested a new fine-tuning framework for ControlNet models(Zhang et al., 2023b) to which the reinforcement learning concept, DDPO(Black et al., 2023), is applied. Our objective is to generate a

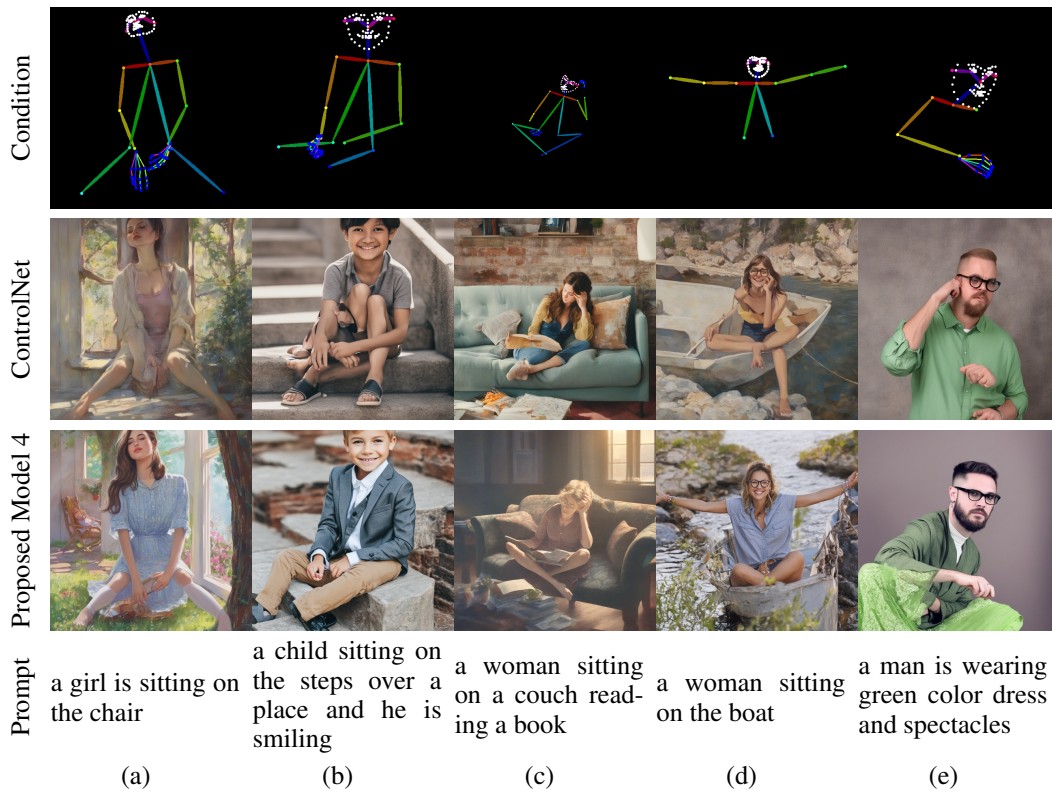

Figure 5: Qualitative Results of OpenPose skeletons with hands and face keypoints. (first row) Skeleton image conditions, (second row) the generated images by the proposed model 3, (third row) the generated images by the proposed model 4, (last row) the input prompts.

human image with more precision in terms of pose. We proposed two kinds of reward functions per the objective: Feature reward function and Keypoints reward function. Through the experiments, we demonstrated that applying our reward function and the framework enhanced the pose accuracy of the models while maintaining its existing ability concerning image quality and text alignment. Specifically, two main points have been checked from the experiments. First, when using the dataset with more details, OpenPose with hands and face(Cao et al., 2017) in our experiment, the pose accuracy was improved compared to the training on the less detailed dataset. Second, both reward functions ensure the enhancement of pose accuracy. Our proposed architecture demonstrates that the application of reinforcement learning to ControlNet(Zhang et al., 2023b) improves the reflection of conditioning images, evidenced by increased pose accuracy in generating human images.

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

## A  TRAINING PROGRESSION

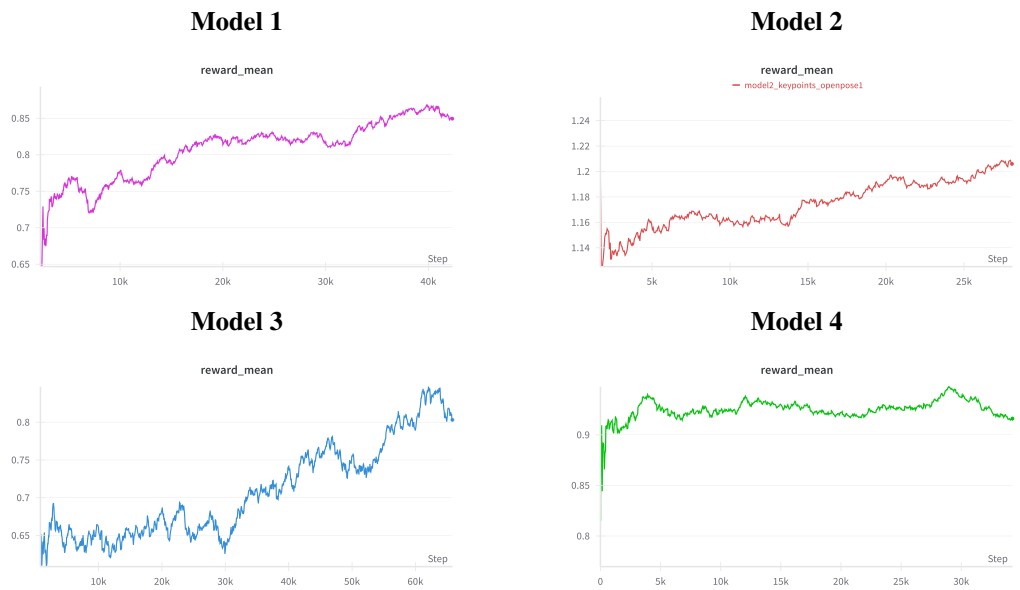

Figure 6: Training Results. Reward changes over training steps. (model 1&3) VLM+Feature Reward, (model 2&4) VLM+KeyPoints Reward

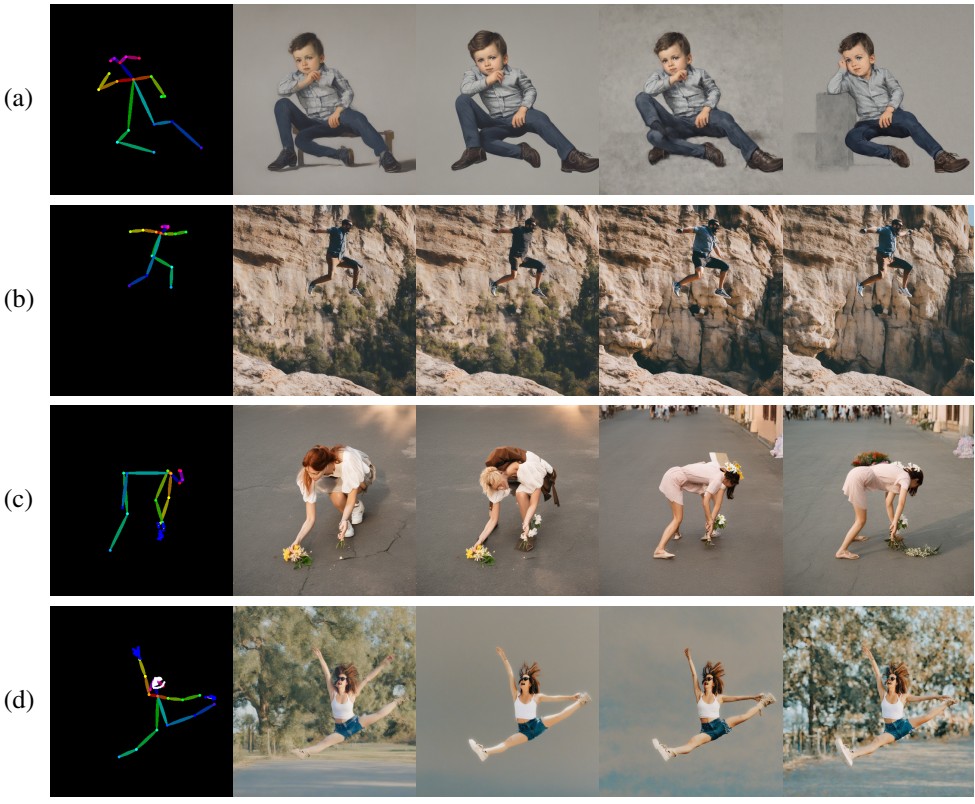

Figure 7: Progression of the models along steps. (first column) Skeleton image conditions, (rest columns) The pose is getting well followed to the conditioning image while training. (row (a) ~ (d)) The generated images by the proposed model 1 ~ 4.

Figure 6 shows that the reward score increases while training, which indicates the models are well following the intention of our reward functions and the framework. We stopped the training when the validation images show the signs of overfitting, such as blurry or unrecognizable background.

Figure 7 shows the progression of the models with regard to the pose accuracy while training. The models end up generating the images following the conditioning images correctly without omissions or additions of the body parts. For example, In Fig. 7-row (a), which is the images of the model 1, extra leg is generated in the earlier steps but it disappears and generates the legs properly following the conditioning images well. In row (b), the model 2 omits left arm at first but the arm appeared in the third image. It generates the left arm correctly at last. In row (c) and (d), the images generated by model 3 and 4, the direction of the body or the arm is following the conditioning images while it was wrong in the earlier steps.

## B    ADDITIONAL SAMPLES

We provide further qualitative comparisons between the baseline models and our proposed models using various pose conditioning images. In these results, we show how our models handle different poses and demonstrate improved accuracy in pose alignment compared to the baseline models.

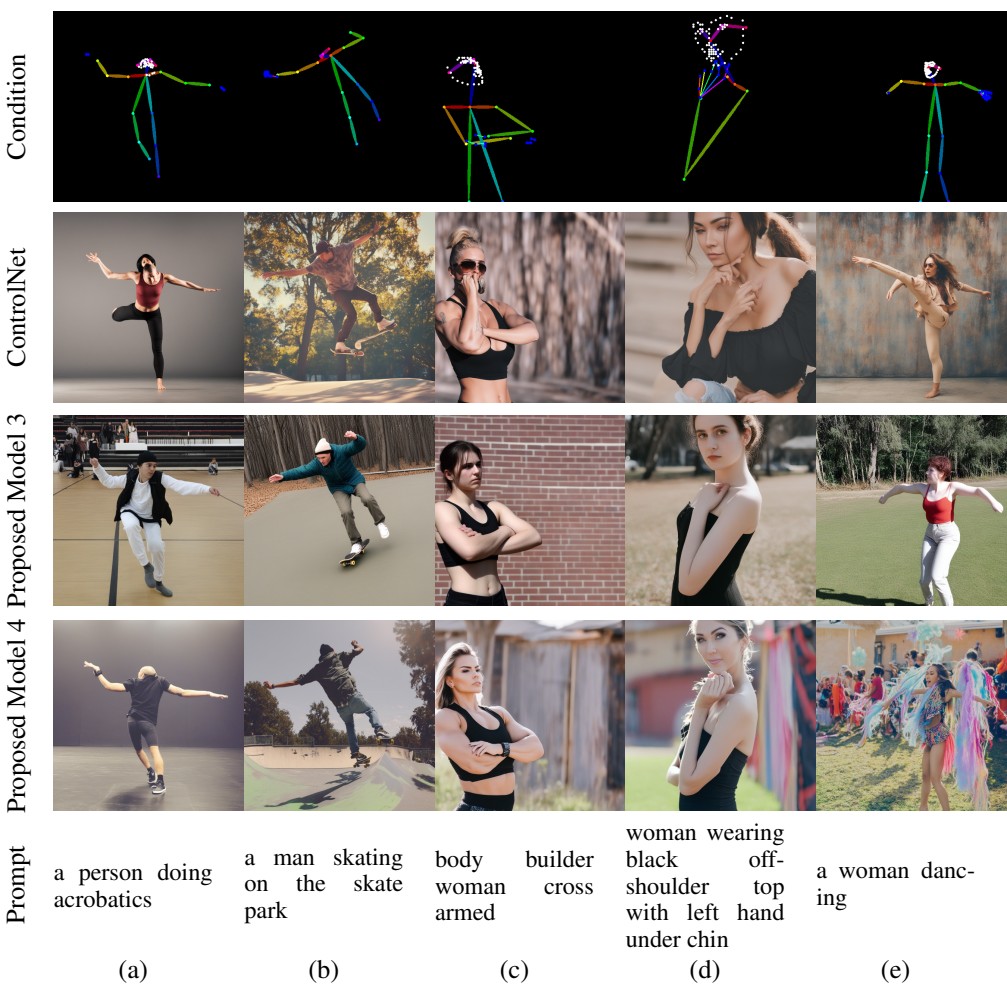

Figure 8: Additional Qualitative Results(w/ hands and face). The results are generated with condition images with hands and face keypoints. (first row) Skeleton image conditions, (second row) the generated images by the baseline model, ControlNet, (third and fourth row) the generated images by our model, the proposed model 3 and 4, (last row) the input prompts.

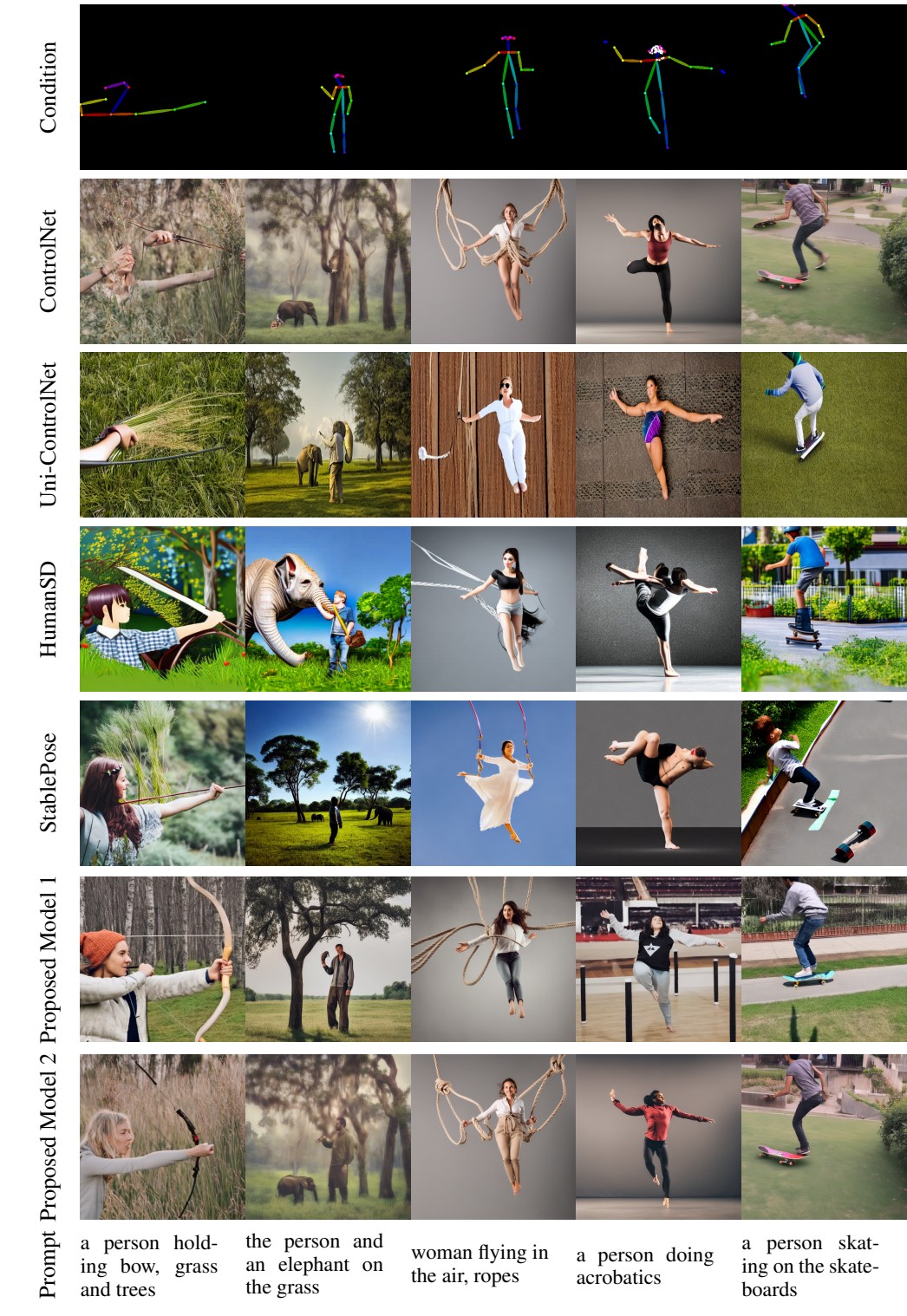

Figure 9: Additional Qualitative Results(w/o hands and face). The results are generated based on conditioning images without hands and face keypoints. (first row) input conditioning images, (second to fifth row) the generated images by the baseline models, ControlNet, Uni-ControlNet, HumanSD and StablePose, (sixth and seventh row) the results from our model 1 and 2, (last row) the input prompts.As inputs, OpenPose images are used for ControlNet, Uni-ControlNet, and ours, and MMPose images are used for HumansSD and StablePose.

