# OpenReview forum: "Improving Human Pose-Conditioned Generation: Fine-tuning ControlNet Models with Reinforcement Learning"
_ICLR.cc/2025/Conference — ICLR 2025 Conference Withdrawn Submission_

### Official Review · Reviewer_w76g · 2024-10-22

**Soundness:** 1
**Presentation:** 2
**Contribution:** 1
**Rating:** 3
**Confidence:** 4

**Summary:**

This paper presents a pose-conditioned human image generation method that combines ControlNet and DDPO. A pose-based reward function is added to DDPO for better pose alignment in the generated image. The authors also collected 21K (human image, pose, caption) triplets for the task.

**Strengths:**

1) The proposed method is explained clearly.

2) The paper is easy to follow.

**Weaknesses:**

1) The motivation for combining DDPO and ControlNet is unclear.
The authors mentioned in L44 that existing methods struggle to generate images that perfectly match the conditioning image pose. What makes the proposed combination of DDPO+ControlNet mitigate or solve this misalignment problem?

2) The second contribution is unexplained in the introduction.
What is the "novel" reward function and its intuition?

3) Missing related work comparison:

   Text2Human: Text-Driven Controllable Human Image Generation. in SIGGRAPH 2022

    HyperHuman: Hyper-Realistic Human Generation with Latent Structural Diffusion. In ICLR 2024

4) The proposed method seems ad-hoc.
Most components in the proposed model have been proposed and introduced in prior work. For example, the model architecture is from ControlNet and the reward-driven training is from DDPO. The proposed method simply combines these two and adds a pose-alignment reward function in DDPO.

5) There is no ablation on removing the proposed pose-based reward function.
Since ControlNet already provides controllability over the pose, how will removing the pose-based reward function affect the results?

6) I suggest the authors provide some visualized examples of failure cases.

7) Objective metrics do not always align well with human perception. To evaluate the image quality, I suggest the authors conduct user studies.

**Questions:**

1) Will the dataset be released?

2) DeepFashion-MultiModal includes ~ 40K (human image, caption, pose) triplets and is a known benchmark. Could the authors provide experiment results on DeepFashion-MultiModal?

3) See Weaknesses for the remaining questions.

---

### Official Review · Reviewer_BFBg · 2024-10-25

**Soundness:** 3
**Presentation:** 2
**Contribution:** 1
**Rating:** 3
**Confidence:** 4

**Summary:**

This paper mainly focuses on generated human body images conditioned on human poses using diffusion models. The authors point out that the commonly used ControlNet cannot generate accurately according to the pose conditions. To this end, the authors propose leveraging the reinforcement learning especially DDPO to train the ControlNet, which is shown to be partially effective given the experiment results.

**Strengths:**

1. The proposed method is simple and easy to implement.
2. The method is clearly described.

**Weaknesses:**

1. The novelty is limited. The utilized DDPO is almost the same as the original one, with the only significant difference being the optimized parameter group.
2. The authors need to reorganize the paper content. For example, Fig.1 and Fig.2 seem similar to each other.
3. As I can understand Tab.3, among all 6 metrics, the proposed method is only better than ControlNet on half of them. I am not sure how this can show the superiority of the proposed method.
4. As for the qualitative comparison, since the authors finetune the pretrained model on extra data, the image quality would certainly be improved as long as the training data quality is high. Besides, the authors should include a simple baseline that finetunes the pretrained ControlNet directly on the training data.
5. The author may want to include HyperHuman [1] as a competitor.
6. It seems the generated results for the proposed model 2 in Fig.9 are still not so satisfactory.

[1] Liu X, Ren J, Siarohin A, et al. Hyperhuman: Hyper-realistic human generation with latent structural diffusion[J]. arXiv preprint arXiv:2310.08579, 2023.

**Questions:**

Please refer to the weaknesses.

---

### Official Review · Reviewer_PomS · 2024-11-03

**Soundness:** 2
**Presentation:** 1
**Contribution:** 2
**Rating:** 3
**Confidence:** 4

**Summary:**

The paper proposes a framework leveraging DDPO to fine-tune a ControlNet for pose-conditioned image generation with two kinds of reward functions (text alignment score plus image similarity/keypoint similarity score).

**Strengths:**

The proposed method seems to be able to generate human images better aligned with input pose conditions and the OKS is improved compared to baseline.

**Weaknesses:**

1. The paper seems to be written in a rushed manner and is hard to follow in some parts. For the proposed reward function, I see no rigorous mathematical definitions. The use of `Model 1, Model 2, Model 3, Model 4` yields a low readability, as I have to refer to Table 2 constantly. It also lacks important details, such as how the reward functions are weighted.

2. Lack of novelty. DDPO has been widely used for fine-tuning diffusion models and is readily integrated in the Huggingface diffusers library. Using it with ControlNet does not constitute a contribution.

3. Weak performance. The proposed method yields lower CLIP scores and cannot improve LPIPS consistently.

4. Unreasonable reward function for keypoint similarity. While cosine similarity can reflect the similarity between two vectors, better metrics, such as OKS, already exist to determine keypoint similarity. I see no sense in using cosine similarity here.

**Questions:**

The work seems incomplete and lacks important details, which cannot be addressed with minor modifications to the original paper, and therefore, my inclination is to reject it. Here are several things to consider

- there are three reward functions: A: VLM-reward function, B: feature similarity, and C: keypoint similarity. They can all be combined instead of doing A+B / A+C combination (as in the paper). There should be some proper weighting between the three terms, which should be shown in the ablation study.

- the comparisons in Table 4 are meaningless because the condition types used are different. The authors should employ the same condition type for different models to make it an apple-to-apple comparison.

---

### Official Review · Reviewer_aHkX · 2024-11-04

**Soundness:** 1
**Presentation:** 1
**Contribution:** 1
**Rating:** 3
**Confidence:** 4

**Summary:**

This paper combines ControlNet and Denoising Diffusion Policy Optimization in human pose conditional generation task. It designs reward function that adds text alignment score and pose similarity score together by using vision language model and pose estimation model. It conducts some experiments comparing previous human pose guided models such as ControlNet and HumanSD and showing higher score on some metrics.

**Strengths:**

This paper focuses on an interesting question, which as great significance to downstreaming research and tasks.

**Weaknesses:**

(1) The presentation of this paper is incomplete, lacking the appropriate structure for a formal paper, with numerous typographical and logical errors, which leads to a reasonable suspicion of LLM involvement in its creation.

(2) II am uncertain about the novelty and advantages of combining ControlNet and DDPO. Could you clarify their unique contributions when used together?

(3) The experiment setup is far from comprehensive. The OKS metric is weak in evaluating human pose estimation, using AP and AR metrics would be more appropriate. Additionally, using LPIPS to assess controlled generation results is quite unusual since generation is expected to be highly diverse.

**Questions:**

See weekness.

---

### Note · Authors · 2024-11-23

**Comment:**

Thank you for all your considerate reviews.
It will be the great help to make more progress on our research.

**Withdrawal Confirmation:**

I have read and agree with the venue's withdrawal policy on behalf of myself and my co-authors.